# The Risk of Cardiovascular Disease According to Chewing Status Could Be Modulated by Healthy Diet in Middle-Aged Koreans

**DOI:** 10.3390/nu14183849

**Published:** 2022-09-17

**Authors:** Hyejin Chun, Jongchul Oh, Miae Doo

**Affiliations:** 1Department of Family Medicine, Ewha Womans University College of Medicine, Seoul 07804, Korea; 2Department of Mathematics, Kunsan National University, Gunsan 54150, Korea; 3Department of Food and Nutrition, Kunsan National University, Gunsan 54150, Korea

**Keywords:** cardiovascular disease, chewing status, Korean Heathy Eating Index, Korean National Health and Nutrition Examination Survey

## Abstract

To assess whether a healthy diet could change the risk of cardiovascular disease (CVD)-related chewing difficulty (CD) in relation to age distribution. In a cross-sectional study of 9411 middle-aged Koreans from the KNHANES VII. In this study, the Framingham 10-year general CVD risk prediction equations and the Korean Heathy Eating index (KHEI) were used to assess the 10-year estimated risk of CVD and dietary quality, respectively. CD was experienced by 16.7% of the total subjects. Among subjects with CD, the 10-year estimated CVD risk was 8.71% of the subjects in the 30–49 years age group and 30.38% of those in the 50–64 years age group, which is a difference of approximately 3.5 times. Regardless of age distribution, the total score of the KHEI in subjects who had CD was significantly lower than in those who had no CD (NCD) (*p* = 0.004 for the 30–49 years age group and *p* < 0.001 for the 50–64 years age group, respectively). Among the subjects with poor KHEI in the 30–49-year age group, the adjusted odds ratio for the 10-year estimated CVD risk of the subjects with CD was 2.204-fold (95% CI = 1.385–3.506) higher using NCD as a reference. The findings showed that dietary quality could modify the risk for CVD according to chewing status.

## 1. Introduction

Globally, cardiovascular disease (CVD) is regarded as the main cause of death. According to the data reported by the World Health Organization (WHO) in 2019, the number of people who died from CVD was approximately 17.9 million, accounting for 32% of all deaths [1]. A similar trend was observed in South Korea, where CVD is the leading cause of death [2]. As the burden due to CVD increases, CVD is recognized as a serious public health problem, and various efforts are being made to reduce it. In line with the trend, it is generally recognized that gender, age, high blood pressure, smoking status, dyslipidemia, diabetes mellitus, and race in some cases are the major risk factors in many previous studies [1,3,4], and as a way to reduce the burden of CVD, it has been reported that identifying high-risk candidates for CVD events from using a combination of those risk factors is necessary [4,5].

The WHO describes that an unhealthy diet is one of the most important behavioral risk factors for CVD [1]. Therefore, reducing salt, consuming more fruit and vegetables, and avoiding excessive use of alcohol are recommended. In 2019, the American College of Cardiology (ACC) and American Heart Association (AHA) recommended that a heathy diet, as a critical impact on CVD and its risk factors, should be consumed by all adults [3,6]. These healthy diets encompass whole grain food, rich in fruits and vegetables, healthy sources of protein (for example, plant protein, fish, fat-free and unprocessed forms of meat or poultry), healthy plant oils, and minimizing the intake of processed foods. A recent review reported that dietary recommendations for CVD emphasized the overall quality of a healthy diet more than specific nutrients or food groups alone [7].

In the past, it was only recognized that chewing was the first process of digestion, mixing with saliva and breaking food into small particles that were swallowed into the esophagus [8]. Recently, chewing status has been associated with various chronic diseases, such as obesity [9,10], metabolic syndrome [11], cognitive function [12,13] and CVD [14,15], as well as nutritional status [16]. These associations are considered to be a health problem that could occur not only due to the deterioration of chewing status due to aging in the elderly population but also in middle age populations.

Although the main risk factors for developing CVD and the 10-year estimated risk of CVD derived from a combination of those were well accepted [4,17] and the association of chewing status with CVD have been reported [14,15], the alteration of association between 10-year estimated risk of CVD and chewing status by dietary quality has not been reported in a large-scale population. Therefore, in this study, the Framingham 10-year general CVD risk prediction equations were used for assessing the 10-year estimated risk of CVD [4], and the Korean Heathy Eating index (KHEI) was used for assessing dietary quality [18]. Subjects were divided into two groups, age groups of 30–49 years and 50–64 years, by considering the data of middle-aged subjects who participated in the Korean National Health and Nutrition Examination Survey VII (KNHANES) [19]. This study examined how the effect of chewing status on the 10-year estimated risk of CVD is influenced by dietary quality in relation to age distribution.

## 2. Subjects and Methods

### 2.1. Data Resources and Subjects Selects

This study used raw data from KNHANES VII (2016–2018), which is a nationwide cross-sectional survey carried out by the Korea Centers for Disease Control and Prevention (KCDC), to investigate the health and nutritional status of Koreans since 1998. It. consisted of a health interview, a health examination, and a dietary survey [19]. From a total of 20,269 participants (11,071 men and 13,198 women) in the raw data from KNHANES VII (2016–2018), 12,080 subjects aged 30–64 years were selected for this study. Subjects with absent or inadequate chewing status and components for 10-year risk estimated CVD data were excluded. Those who reported an implausible daily total energy consumption of ≤500 kcal or ≥5000 kcal were excluded. Finally, the remaining 9411 subjects were selected for this study. All participants signed the provided written informed consent, and the survey protocol was approved by the Institutional Review Board of the KCDC [2018-01-03-P-A].

### 2.2. Measurements

#### 2.2.1. Chewing Status

In the health interviews, chewing status was collected by trained specialists using questionnaires. Responses were given on a 5-point scale to the question of whether or not subjects are currently experiencing discomfort in chewing due to problems in the mouth, such as teeth, dentures, and gums. If dentures were used, the subjects were asked to answer while wearing their dentures. Subjects were divided into “chewing difficulty (CD)” for “very uncomfortable” and “uncomfortable” and “no chewing (NCD)” for “moderate”, “not uncomfortable”, and “not at all uncomfortable” according to the question of mastication discomfort [20].

#### 2.2.2. Sociodemographic Characteristics

From the health interview, sociodemographic characteristics such as educational level, household income, regular status of alcohol consumption, aerobic physical activity, and perceived stress were used for this study. The subjects were classified into two groups: “≤middle school” or “≥high school” according to education level and “low” or “high” according to household income based on their median income. According to the regular status of alcohol consumption during the past year, subjects were divided into “yes” or “no”. Aerobic physical activity was classified as “yes” or “no” based on activity during the previous week. Perceived stress was divided “low” or “high” according to the degree to which the subjects perceived stress.

#### 2.2.3. Korean Health Eating Index

The Korea Disease Control and Prevention Agency and the Korean Nutrition Society developed the KHEI to identify comprehensive diet quality among Korean adults [18]. It comprises 3 categories (adequacy, moderation, and balance) and 14 components. The adequacy category (8 components) assesses whether food or nutrient consumption is suitable for dietary guidelines for Korean adults or KDRIs. The moderation category (3 components) assesses restricted food or nutrient consumption. The final category, the balance category, includes 3 components. The overall scores possible on this scale ranged from 0 to 100, with higher scores indicating healthy eating.

#### 2.2.4. Factors of Cardiovascular Disease Risk (CVD) and Definitions of 10-Year Estimated CVD Risk

The data from the health interviews and health examination in the KNHANE and the 10-year estimated CVD risk were calculated. Age, sex, total cholesterol (TC), high dense lipoprotein-cholesterol (HDL-C), systolic blood pressure (SBP), hypertension treatment, smoking status, and diagnosis of diabetes mellitus were used in the functions to calculate the 10-year estimated CVD risk [4]. After fasting overnight, blood samples were collected and measured for TC and HDL-C using an automatic analyzer 7600 (Hitachi Ltd., Tokyo, Japan). BP was measured twice using a mercury manometer, and the average was used for the analysis. The smoking status, medical history of hypertension treatment and diagnosis of diabetes mellitus were self-reported based on the health interview.

The 10-year estimated CVD risk was calculated using the following previously reported functions [4].

For men, 1 − 0.88936^exp (Σ^*^β^*
^X−23.9802)^.

For women, 1 − 0.95012^exp (Σ^*^β^*
^X−26.1931)^, where *β* is the regression coefficient and X is the level for each risk factor (Table 1).

In this study, intermediate and high risk for 10-year estimated CVD was defined as ≥10% [21].

### 2.3. Statistical Analyses

All statistical analyses were carried out using SPSS (version 27.0; IBM Corp., Armonk, NY, USA) software for Windows. Weighed complex sampling was applied in all analyses to reflect estimates of the entire Korean population. Continuous variables are presented as estimated means (95% confidence intervals), and categorical variables are presented as weighted percentages (standard errors). Pearson’s chi-square test and t test were used to identify the sociodemographic characteristics, ten-year estimated CVD risk and components, and differences in KHEI according to chewing status and age distribution. To identify whether the 10-year estimated CVD risk according to chewing status and age distribution is affected by dietary quality, we used multinomial logistic regression after adjustment for covariates. As covariates, socioeconomic factors (gender, education level, household income) and health-related factors (aerobic physical activity, regular drinker, and perceived stress) were adjusted. After being stratified by the median KHEI, the adjusted odds ratios for 10-year estimated CVD risk according to chewing status and age distribution were calculated in reference to no chewing difficulty (NCD) using multinomial logistic regression. In all analyses, statistical significance was defined as a *p* value < 0.05.

## 3. Results

According to chewing difficulty and age distribution, the general characteristics are shown in Table 2. In this study, 10.4% of subjects aged 30–49 and 25.9% of those aged 50–64 experienced chewing difficulty (0.5% and 0.8% for SE). Regardless of age distribution, subjects who experienced chewing difficulty had a significantly higher proportion of men (*p* = 0.044 for age group of 30–49 years and *p* = 0.008 for age group of 50–64 years), lower educational level (*p* < 0.001 for both), lower household income (*p* < 0.001 for both), and higher perceived stress (*p* < 0.001 for both) than those who did not experience chewing difficulty. Aerobic physical activity according to chewing difficulty differed in the age group of 50–64 years (*p* = 0.001). However, there were no differences in regular drinkers by chewing difficulty and age distribution.

Table 3 and Figure 1 present the 10-year estimated CVD risk and its components and the odds ratios (ORs) and 95% confidence intervals (95% CIs) for 10-year estimated CVD risk according to chewing status and age distribution. Among the risk factors for calculating the 10-year estimated CVD risk, age and smoking status were significantly different by chewing status, regardless of age distribution (*p* < 0.001 for all). In the age group of 30–49 years, there were significant differences observed in TC (*p* = 0.026) and HDL-C (*p* = 0.013). On the other hand, the SBP in the age group of 50–64 years showed a significant difference (*p* = 0.006). Chewing difficulty was associated with 10-year estimated CVD risk. In other words, the subjects with chewing difficulty showed a higher 10-year estimated CVD risk than those with no chewing difficulty (*p* = 0.003 for both). Interestingly, the 10-year estimated CVD risk among subjects who experienced chewing difficulty was 8.71% in the age group of 30–49 years but 30.38% in the age group of 50–64 years. The proportion of the intermediate and high for 10-year estimated CVD risk, which was defined as ≥10%, by chewing status were different in the age group of both 30–49 years and 50–64 years. Among the subjects with chewing difficulty, 23.1% of the subjects were in the age group of 30–49 years, whereas 65.4% of those in the age group of 30–49 years accounted for the intermediate and high 10-year estimated CVD risk (*p* < 0.001 for the age group of 30–49 years and *p* = 0.002 for the age group of 50–64 years). Significant differences were observed in the odds ratio for the 10-year estimated CVD risk by chewing status both in the age group 30–49 years and 50–64 years. The ORs for the 10-year estimated CVD risk of the subjects with chewing difficulty were 1.590-fold (95% CI = 1.245–2.033) higher in the age group of 30–49 years and 1.374 (95% CI = 1.155–1.637) in the age group of 50–64 years using no chewing difficulty as a reference.

Differences in the KHEI according to chewing status and age distribution are presented in Table 4. The total score of the KHEI was significantly lower in subjects who had chewing difficulty than in those who had no chewing difficulty, regardless of age distribution (59.42 vs. 61.64, *p* = 0.004 for the 30–49 years age group and 64.56 vs. 67.29, *p* < 0.001 for the 50–64 years age group, respectively). Among adequacy in three categories, the subjects with chewing difficulty showed lower scores for “total fruit intake (1.96 vs. 1.55, *p* < 0.001 for age group of 30–49 years and 2.33 vs. 2.75, *p* < 0.001 for age group of 50–64 years, respectively)”, “fresh fruit intake (1.66 vs. 2.20, *p* < 0.001 for age group of 30–49 years and 2.62 vs. 2.99, *p* < 0.001 for age group of 50–64 years, respectively)”, and “meat, fish, eggs and beans intake (7.14 vs. 7.59, *p* = 0.006 for age group of 30–49 years and 6.31 vs. 7.02, *p* < 0.001 for age group of 50–64 years, respectively)” than the subjects with no chewing difficulty, regardless of age distribution. The subjects with chewing difficulty among the age group of 50–64 years showed lower scores for “have breakfast (8.40 vs. 8.07, *p* = 0.019)”, “total vegetables intake (2.99 vs. 2.62, *p* = 0.002)”, “vegetable intake excluding Kimchi and pickled vegetables intake (3.44 vs. 3.19, *p* < 0.001)”, and “milk and milk products intake (3.37 vs. 3.00, *p* = 0.043) compared to those with no chewing difficulty. However, among the moderation category, the “percentage of energy from saturated fatty acids” was lower in the subjects with chewing difficulty than in those with chewing difficulty in the age group of 50–64 years (8.08 vs. 8.37, *p* = 0.029), which was contradictory.

To identify whether there was a difference in the effects of chewing status and age distribution on 10-year estimated CVD risk according to total KHEI score, multinomial logistic regression analysis was used after being stratified into groups by the median total KHEI score, as shown in Figure 2. A significant difference in the adjusted odds ratio for 10-year estimated CVD risk in subjects with poor KHEI scores was observed according to chewing status only among the age group of 30–49 years. Among the subjects with poor KHEI scores in the 30–49 years age group, the adjusted OR for the 10-year estimated CVD risk of the subjects with chewing difficulty was 2.204-fold (95% CI = 1.385–3.506) higher using no chewing difficulty as a reference. However, there was no observed 10-year estimated CVD risk according to chewing status among the subjects with good KHEI scores in the 30–49-year age group. In the age group 50–64 years, the adjusted OR for the 10-year estimated CVD risk by chewing status was not different, regardless of the total KHEI score.

## 4. Discussion

Using nationally representative data of the middle-aged Korean population from KNANES, this study examined whether dietary quality influenced the 10-year estimated CVD risk in relation to chewing status and age distribution. According to the chewing status and age distribution, the 10-year estimated CVD risk and total score of the KHEI significantly differed. Additionally, the adjusted odds ratios for 10-year estimated CVD risk were affected by chewing status and KHEI only in the age group of 30–49 years.

In the middle-aged population in this study, the proportion of subjects with chewing difficulty was 16.7%, which was approximately 40.5% lower than that of the elderly population of 41.2% in a previous study [20]. As with a previous study on the inverse association between age and chewing ability [20,22], subjects aged 50–64 years had more chewing difficulty than those aged 30–49 years. Consistent with the results of other studies, although conducted in the middle-aged population in this study [23,24], chewing status was negatively associated with sociodemographic variables, including lower educational level and household income and higher perceived stress in both the 30–49 years and 50–64 years age groups.

Several studies, not in the elderly population, have reported associations of chewing status with risk factors for CVD such as TC, HDL-C, SBP, and smoking status etc. [11,14,25]. Rangé et al. considered that good chewing status as prerequisite for high level of cardiovascular health [14]. Observational study has shown chewing status was associated with blood pressure only middle-aged adult [25]. From the findings of this study, it is reported for the first time that poor chewing status was associated with an increased 10-year estimated CVD risk. Additionally, the results of this study showed that the 10-year estimated CVD risk among subjects who experienced chewing difficulty was 8.71% and 30.38% in the age groups of 30–49 years and 50–64 years, respectively, which is a difference of approximately 3.5 times. These findings make it difficult to fully explain the association of chewing status with CVD risk in relation to age distribution, but these findings could be explained by a decline in chewing ability with aging and the increase in major risk factors for CVD with aging [4,12].

Interestingly, after adjustment for gender, education level, household income aerobic physical activity, regular drinker, and perceived stress, the subjects with chewing difficulty in the age group of 30–49 years showed a difference in the 10-year estimated CVD risk according to total KHEI scores. Among the subjects with low total KHEI scores, to identify the healthy quality of diet used, chewing difficulty was associated with a higher 10-year estimated CVD risk, whereas higher risk was not shown among those with high total KHEI scores. However, in the age group of 50–64 years, no difference was shown in the association of chewing status with higher 10-year estimated CVD risk, regardless of total KHEI scores. As mentioned in the Introduction, having heathy quality of diet is one of the critical ways to reduce and manage CVD risks [1,3,6,7], and good chewing ability is related to high all nutrients intake and high food variety [16]. Consistent with the current findings, maintaining healthy quality even with poor chewing ability might help reduce the risk of CVD.

This study demonstrated that chewing status may influence 10-year estimated CVD risk in a middle-aged population and that this association is modified by dietary quality only in the age group of 30–49 years. However, this study has several limitations. First, the results of this study could not explain the causal relationships between those associations because a cross-sectional design was used. Second, among the major risk factors for CVD, current smoking status, medical history of hypertension treatment and diagnosis of DM were collected by self-report, which might lead to inaccuracies in the calculation for the 10-year estimated CVD risk. Finally, although it was reported to provide a more accurate prediction regardless of the gender in Korean population [26], the Framingham 10-year general CVD risk prediction equation is not applicable to all populations due to differences by age or race.

## 5. Conclusions

This study demonstrated that chewing status is related to the 10-year estimated CVD risk using large-scale data of the middle-aged population from the KNHANES. These associations are affected by the KHEI as dietary quality only in the age group of 30–49 years. The findings from our study suggest that there is a difference in the 10-year estimated CVD risk by age distribution, and poor quality of diet might act as a risk factor for CVD risk in the age group of 30–49 years. Therefore, even a relatively younger middle-aged population with a low 10-year estimated CVD risk should need to control the quality of diet in advance along with the management of major risk factors for CVD.

## Figures and Tables

**Figure 1 nutrients-14-03849-f001:**
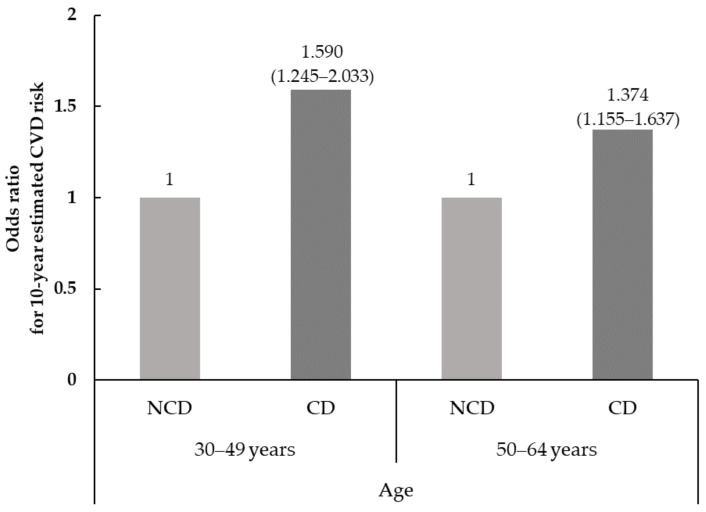
The odds ratios (ORs) and 95% confidence intervals (95% CIs) for 10-year estimated CVD risk according to chewing status. CD; Chewing difficulty, NCD; No chewing difficulty. The odds ratios were calculated in reference to no chewing difficulty (NCD) using logistic regression analysis.

**Figure 2 nutrients-14-03849-f002:**
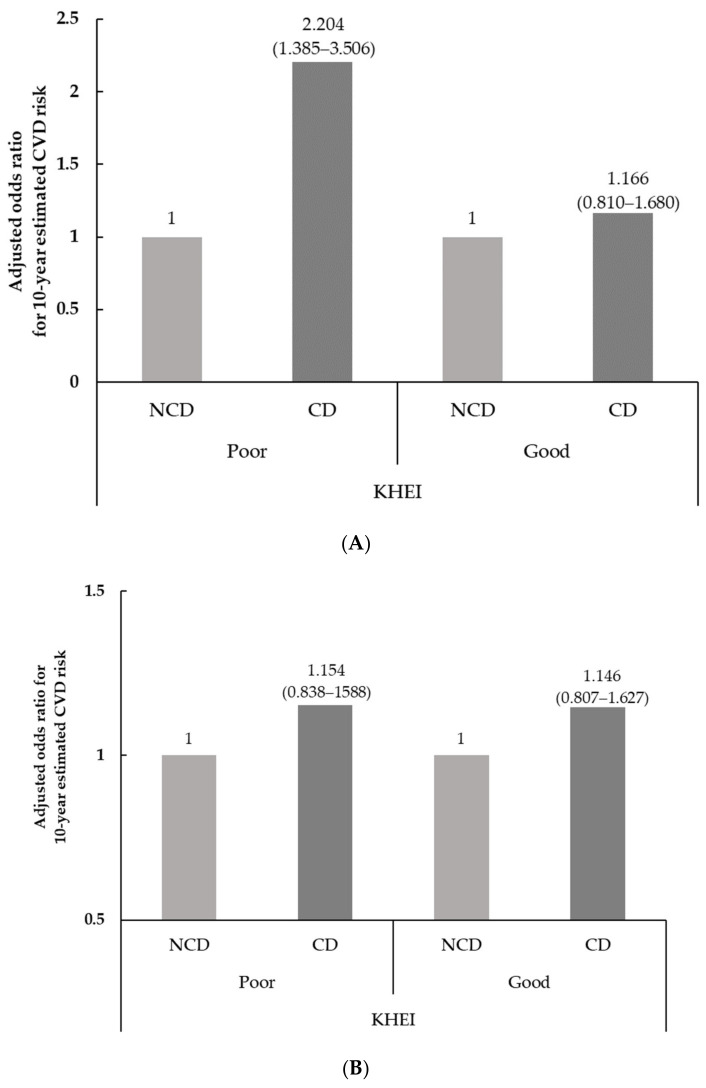
The adjusted odds ratios (ORs) and 95% confidence intervals (95% CIs) for 10-year estimated CVD risk according to chewing status and KHEI (Korean heathy eating index). CD; Chewing difficulty, NCD; No chewing difficulty, KHEI; Korean Healthy Eating Index. (**A**) age group of 30–49 years; (**B**) age group of 50–64 years. After being divided by the media score of the KHEI, the adjusted odds ratios were calculated in reference to no chewing difficulty (NCD) using multinomial logistic regression after gender, education level, household income, regular drinker, aerobic physical activity, and perceived stress.

**Table 1 nutrients-14-03849-t001:** Regression coefficients and hazard ratios for risk factors.

Risk Factors	*β*	*P* Value	HR	95% CI
Men				
Log of Age	3.06117	<0.001	21.35	(14.03–32.48)
Log of total cholesterol	1.12370	<0.001	3.08	(2.05–4.62)
Log of HDL cholesterol	−0.93263	<0.001	0.39	(0.30–0.52)
Log of SBP if not treated	1.93303	<0.001	6.91	(3.91–12.20)
Log of SBP if treated	1.99881	<0.001	7.38	(4.22–12.92)
Smoking status	0.65451	<0.001	1.92	(1.65–2.24)
Diabetes mellitus	0.57367	<0.001	1.78	(1.43–2.20)
Women				
Log of Age	2.32888	<0.001	10.27	(5.65–18.64)
Log of total cholesterol	1.20904	<0.001	3.35	(2.00–5.62)
Log of HDL cholesterol	−0.70833	<0.001	0.49	(0.35–0.69)
Log of SBP if not treated	2.76157	<0.001	15.82	(7.86–31.87)
Log of SBP if treated	2.82263	<0.001	16.82	(8.46–33.46)
Smoking status	0.52873	<0.001	1.70	(1.40–2.06)
Diabetes mellitus	0.69154	<0.001	2.00	(1.49–2.67)

HR; Hazard Ratio, CI; Confidence interval, HDL; High dense lipoprotein, SBP; Systolic blood pressure.

**Table 2 nutrients-14-03849-t002:** General characteristics by chewing status and age distribution in middle-aged Koreans.

	30~49 Years (*n* = 5159)			50~64 Years (*n* = 4252)		
	CD (*n* = 541)	NCD (*n* = 4618)	*p* Value *	CD (*n* = 1129)	NCD (*n* = 3123)	*p* Value *
Gender, men	55.5 (2.3)	50.0 (0.7)	0.044	53.0 (1.6)	47.7 (1.0)	0.008
Education level, ≥high school	9.9 (0.5)	90.1 (0.5)	<0.001	20.5 (0.9)	79.5 (0.9)	<0.001
Household income, high	58.0 (2.5)	72.2 (1.0)	<0.001	57.2 (1.9)	70.8 (1.2)	<0.001
Regular drinker, yes	67.5 (2.5)	66.6 (0.8)	0.737	56.9 (1.7)	53.7 (1.0)	0.106
Aerobic physical activity, yes	44.9 (2.6)	49.5 (0.9)	0.102	37.5 (1.8)	44.4 (1.1)	0.001
Perceived stress, low	60.7 (2.3)	70.3 (0.8)	<0.001	67.8 (1.7)	79.5 (0.9)	<0.001

Data are presented as weighted percentages (standard errors) for categorical variables, unless otherwise stated. CD; Chewing difficulty, NCD; No chewing difficulty * *p* value between chewing status x2-test or *t* test.

**Table 3 nutrients-14-03849-t003:** Ten-year estimated CVD risk and its components by chewing status and age distribution in middle aged Korean.

	30~49 Years (*n =* 5159)			50~64 Year (*n =* 4252)		
	CD	NCD	*p* Value *	CD	NCD	*p* Value *
	(*n =* 541)	(*n =* 4618)		(*n =* 1129)	(*n =* 3123)	
Age, year	41.54 (40.99–42.10)	39.84 (39.61–40.07)	<0.001	56.99 (56.70–57.29)	56.19 (56.00–56.37)	<0.001
TC, mg/dL	193.00 (189.81–196.19)	196.83 (195.60–198.05)	0.026	198.53 (195.75–201.30)	199.60 (197.89–201.30)	0.504
HDL-C, mg/dL	50.22 (49.05–51.40)	51.85 (51.41–52.29)	0.013	50.37 (49.44–51.29)	50.69 (50.12–51.26)	0.562
SBP, mmHg	114.04 (112.60–115.47)	113.24 (112.71–113.76)	0.279	122.52 (121.29–123.76)	120.55 (119.84–121.26)	0.006
BP treatment, %	5.3 (1.1)	4.6 (0.4)	0.517	25.3 (1.6)	24.1 (0.8)	0.503
Smoking status, %	36.4 (2.5)	23.0 (0.8)	<0.001	29.2 (1.8)	17.3 (0.9)	<0.001
DM diagnosis, %	2.3 (0.7)	2.0 (0.2)	0.690	12.1 (1.2)	9.9 (0.6)	0.093
10-year estimated CVD risk, %	8.71 (7.54–9.87)	6.84 (6.46–7.22)	0.003	30.38 (28.12–32.65)	26.42 (25.31–27.54)	0.003
Intermediate/high for CVD risk **	23.1 (2.1)	15.9 (0.6)	<0.001	65.4 (1.7)	57.9 (1.0)	0.002

Data are presented as estimated means (95% confidence intervals) for continuous variables or weighted percentages (standard errors) for categorical variables, unless otherwise stated. CD; Chewing difficulty, NCD; No chewing difficulty, TC; Total cholesterol, HDL-C; High dense lipoprotein-cholesterol, SBP; Systolic blood pressure, BP; blood pressure, DM; Diabetes mellitus, 10-year estimated CVD risk; 10-year estimated cardiovascular disease risk. * *P* values between chewing status using the x2-test or *t* test. ** Intermediates/high CVD risk were defined as ≥10%.

**Table 4 nutrients-14-03849-t004:** Korean healthy eating index by chewing status and age distribution in middle aged Korean.

	Scoring	30~49 Years (*n* = 5159)			50~64 Years (*n* = 4252)		
		CD (*n* = 541)	NCD (*n* = 4618)	*p* Value *	CD (*n* = 1129)	NCD (*n* = 3123)	*p* Value *
Have breakfast	0–10	5.98 (5.55–6.41)	6.26 (6.10–6.42)	0.217	8.07 (7.83–8.31)	8.40 (8.25–8.55)	0.019
Mixed grains intake	0–5	1.56 (1.37–1.74)	1.65 (1.58–1.73)	0.318	2.15 (2.00–2.30)	2.29 (2.19–2.38)	0.105
Total fruits intake	0–5	1.55 (1.34–1.76)	1.96 (1.88–2.04)	<0.001	2.33 (2.17–2.49)	2.75 (2.65–2.85)	<0.001
Fresh fruits intake	0–5	1.66 (1.43–1.90)	2.20 (2.11–2.29)	<0.001	2.62 (2.45–2.79)	2.99 (2.88–3.09)	<0.001
Total vegetables intake	0–5	3.52 (3.38–3.66)	3.50 (3.45–3.55)	0.803	3.60 (3.49–3.72)	3.80 (3.74–3.87)	0.002
Vegetables intake excluding Kimchi and pickled vegetables intake	0–5	3.14 (2.98–3.31)	3.22 (3.16–3.27)	0.416	3.19 (3.07–3.32)	3.44 (3.37–3.51)	<0.001
Meat, fish, eggs and beans intake	0–10	7.14 (6.84–7.45)	7.59 (7.49–7.69)	0.006	6.31 (6.08–6.54)	7.02 (6.88–7.16)	<0.001
Milk and milk products intake	0–10	3.20 (2.75–3.64)	3.34 (3.19–3.50)	0.535	3.00 (2.68–3.31)	3.37 (3.16–3.58)	0.043
Percentage of energy from saturated fatty acid	0–10	6.79 (6.39–7.19)	6.78 (6.64–6.93)	0.957	8.37 (8.15–8.59)	8.08 (7.92–8.23)	0.029
Sodium intake	0–10	6.32 (6.00–6.64)	6.31 (6.19–6.43)	0.946	6.97 (6.71–7.22)	6.68 (6.53–6.82)	0.050
Percentage of energy from Sweets and beverages	0–10	9.08 (8.99–9.16)	8.82 (8.54–9.10)	0.079	9.29 (9.15–9.44)	9.44 (9.37–9.52)	0.066
Percentage of energy from carbohydrate	0–5	2.95 (2.75–3.16)	2.91 (2.85–2.99)	0.696	2.30 (2.15–2.45)	2.46 (2.38–2.55)	0.068
Percentage of energy from fat	0–5	3.70 (3.51–3.90)	3.68 (3.61–3.75)	0.851	3.26 (3.11–3.41)	3.41 (3.32–3.50)	0.093
Energy Intake	0–5	3.09 (2.89–3.29)	3.16 (3.09–3.23)	0.518	3.12 (2.96–3.27)	3.16 (3.07–3.25)	0.568
Total score of KHEI		59.42 (57.98–60.87)	61.64 (61.13–62.14)	0.004	64.56 (63.63–65.50)	67.29 (66.71–67.87)	<0.001

Data are presented as estimated means (95% confidence intervals). CD; Chewing difficulty, NCD; No chewing difficulty, KHEI; Korean Healthy Eating Index. * *P* values between status using the x2-test or *t* test.

## Data Availability

The data used in this study are available from the Korea Centers for Disease Control and Prevention, following webpage: https://knhanes.cdc.go.kr/ (accessed on 10 June 2022).

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
