# Peer review of "The Risk of Cardiovascular Disease According to Chewing Status Could Be Modulated by Healthy Diet in Middle-Aged Koreans"

_nutrients, 2022, doi:10.3390/nu14183849_

Round 1

Reviewer 1 Report

This is a high quality study investigating the influence of chewing status and healthy diet on CVD risk. The authors have demonstrated that chewing disorders are associated with higher CVD risk and poor dietary quality especially in the age group 30-49 years in a large Korean cohort.

In the Abstract section line 13 it saysIn In a cross-sectional study of 9,411...”, please remove one In.

Please, state brief conclusion in the Abstract section.

In the Method section please clarify which questionnaires were used for assessing chewing difficulties.

In Table 1 Abbreviations are missing.

Author Response

Manuscript ID: nutrients-1893393
Type of manuscript: Article 
Title: The Risk of Cardiovascular Disease according to Chewing status could be Modulated by Healthy diet in middle -aged Koreans.
Authors: Hyejin Chun, Jongchul Oh, Miae Doo *

----------------------------------------------------------------------------------------

Reviewer # 1

This is a high quality study investigating the influence of chewing status and healthy diet on CVD risk. The authors have demonstrated that chewing disorders are associated with higher CVD risk and poor dietary quality especially in the age group 30-49 years in a large Korean cohort.

We appreciate the reviewer for the careful reading and description of our manuscript with the valuable comments. We worked to the best of my abilities to revise the issues the reviewer pointed out.

In the Abstract section line 13 it says „In In a cross-sectional study of 9,411...”, please remove one In.

It has been removed on the comments

Please, state brief conclusion in the Abstract section.

It has been added on the comments as follows: line 23~24.

“The findings showed that dietary quality could modify the risk for CVD according to chewing status.”

In the Method section please clarify which questionnaires were used for assessing chewing difficulties.

It has been revised based on the comments as follows: line 86~92

“Responses were given on a 5-point scale to the question of whether or not subjects are currently experiencing discomfort in chewing due to problems in the mouth, such as teeth, dentures, and gums. If dentures were used, the subjects were asked to answer while wearing their dentures. Subjects were divided into “chewing difficulty (CD)” for “very uncomfortable” and “uncomfortable” and “no chewing (NCD)” for “moderate”, “not uncomfortable”, and “not at all uncomfortable” according to the question of mastication discomfort [20].”

In Table 1 Abbreviations are missing.

It has been added on the comments

Reviewer 2 Report

The aim of this manuscript is to investigate the effect of chewing difficulty (CD) and diet habit in the risk of 10 year cardiovascular disease (CVD) in different age groups in Korean population. More than 20,000 participants were involved in this cross-sectional survey, and the author found that 10-year CVD risk was 3.5 time higher in participant with CD at older age. They also found that for participant with poor diet habit, the 10-year CVD risk was significantly higher than healthy diet habit subject at younger age.

1       1. For Table 3, Smoking status%, TC, HDL and TC showed statistic significant but not clinical significant different between CD and NCD groups in both age range. Therefore, these confounding factors should be adjusted when compare the 10-year CVD risk between these groups.

2       2. Same adjustment should be done for Figure 1

3      3.  Further adjustment should be done for Figure 2. For example, Smoking status%, HDL, TC should be adjusted between CD and NCD as it might affect the 10-year CVD risk.

4      4. More detailed chowing difficulties scale should be provided.

Author Response

Manuscript ID: nutrients-1893393
Type of manuscript: Article 
Title: The Risk of Cardiovascular Disease according to Chewing status could be Modulated by Healthy diet in middle -aged Koreans.
Authors: Hyejin Chun, Jongchul Oh, Miae Doo *

----------------------------------------------------------------------------------------

Reviewer # 2

The aim of this manuscript is to investigate the effect of chewing difficulty (CD) and diet habit in the risk of 10 year cardiovascular disease (CVD) in different age groups in Korean population. More than 20,000 participants were involved in this cross-sectional survey, and the author found that 10-year CVD risk was 3.5 time higher in participant with CD at older age. They also found that for participant with poor diet habit, the 10-year CVD risk was significantly higher than healthy diet habit subject at younger age.

We appreciate the reviewer for the careful reading and description of our manuscript with the valuable comments. We worked to the best of my abilities to revise the issues the reviewer pointed out.

  1. For Table 3, Smoking status%, TC, HDL and TC showed statistic significant but not clinical significant different between CD and NCD groups in both age range. Therefore, these confounding factors should be adjusted when compare the 10-year CVD risk between these groups.

As you pointed out, significant differences in smoking status, TC, HDL-C, SBP according to chewing status and age distribution were observed in the Table 3. However, those were used to calculate the 10-year estimated CVD risk. The 10-year estimated CVD risk was reflected TC, smoking status, HDL, etc.

  1. Same adjustment should be done for Figure 1

Fig 1 is the ddd ratios using the 10-year estimated CVD risk as in Table 3.

  1. Further adjustment should be done for Figure 2. For example, Smoking status%, HDL, TC should be adjusted between CD and NCD as it might affect the 10-year CVD risk.

Thank you for comments. In the Fig 2, the 10-year estimated CVD risk was calculated smoking status, TC, HDL-C, etc. So those could not adjust the 10-year estimated CVD risk.

  1. More detailed chowing difficulties scale should be provided.

It has been revised based on the comments as follows: line 86~92

“Responses were given on a 5-point scale to the question of whether or not subjects are currently experiencing discomfort in chewing due to problems in the mouth, such as teeth, dentures, and gums. If dentures were used, the subjects were asked to answer while wearing their dentures. Subjects were divided into “chewing difficulty (CD)” for “very uncomfortable” and “uncomfortable” and “no chewing (NCD)” for “moderate”, “not uncomfortable”, and “not at all uncomfortable” according to the question of mastication discomfort [20].”

Round 2

Reviewer 2 Report

The author response all the comments with some improvement of this manuscript. However, the aim of this manuscript are still not clear or has only some significant to this field.

1. The association between CD and different CVD risk factors (eg. HLD, LDL, BP)  should be emphasized as showed in Table 3.  

2. Figure 1, should be the "KHEI" replaced with "Age"?

Author Response

Reviewer # 2

The author response all the comments with some improvement of this manuscript. However, the aim of this manuscript are still not clear or has only some significant to this field.

  1. The association between CD and different CVD risk factors (eg. HLD, LDL, BP)  should be emphasized as showed in Table 3.  

We appreciate the constructive and very helpful comments. Like your comment, we agreed that the chewing status was associated with various factors for CVD risk Therefore, we rewritten DISCUSSION, as follow line261-264.

“Several studies, not in the elderly population, have reported associations of chewing status with risk factors for CVD such as TC, HDL-C, SBP, and smoking status and so on [11, 14, 24]. Rangé et al considered that good chewing status as prerequisite for high level of cardiovascular health [14]. Observational study has shown chewing status was associated with blood pressure only middle-aged adult [24].”

  1. Figure 1, should be the "KHEI" replaced with "Age"?

It has been revised based on the comments